# Effect of thermal control of dry fomites on regulating the survival of human pathogenic bacteria responsible for nosocomial infections

Tomoko Shimoda[1☯], Torahiko Okubo[2☯], Yoshiki Enoeda[1], Rika Yano[1], Shinji Nakamura[3], Jeewan Thapa[2], Hiroyuki Yamaguchi[2]*

1 Department of Fundamental Nursing, Faculty of Health Sciences, Hokkaido University, Sapporo, Japan,
2 Department of Medical Laboratory Science, Faculty of Health Sciences, Hokkaido University, Sapporo, Japan, 3 Laboratory of Morphology and Image Analysis, Biomedical Research Center, Juntendo University Graduate School of Medicine, Tokyo, Japan

☯ These authors contributed equally to this work.
* hiroyuki@med.hokudai.ac.jp

**Data Availability Statement:** All relevant data are within the manuscript and its Supporting Information files.

## Abstract

We monitored the survival of human pathogenic bacteria [*Escherichia coli* (ATCC), extended-spectrum β-lactamase-producing *E. coli* (Clinical isolate), New Delhi metallo-β-lactamase-producing *E. coli* (clinical isolate), *Staphylococcus aureus* (ATCC)] on dry materials (vinyl chloride, aluminum, plastic, stainless steel) at distinct temperatures (room temperature or 15˚C–37˚C). These bacteria favored a lower temperature for their prolonged survival on the dry fomites, regardless of the material type. Interestingly, when mixed with *S. aureus*, *E. coli* survived for a longer time at a lower temperature. Cardiolipin, which can promote the survival of *S. aureus* in harsh environments, had no effect on maintaining the survival of *E. coli*. Although the trends remained unchanged, adjusting the humidity from 40% to 60% affected the survival of bacteria on dry surfaces. Scanning electron microscopic analysis revealed no morphological differences in these bacteria immediately before or after one day of dry conditions. In addition, ATP assessment, a method used to visualize high-touch surfaces in hospitals, was not effective at monitoring bacterial dynamics. A specialized handrail device fitted with a heater, which was maintained at normal human body core temperature, successfully prohibited the prolonged survival of bacteria [*Enterococcus faecalis* (ATCC), *E. coli* (ATCC), *Pseudomonas aeruginosa* (ATCC), *S. aureus* (ATCC), *Acinetobacter baumannii* (clinical isolate), and *Serratia marcescens* (clinical isolate)], with the exception of spore-forming *Bacillus subtilis* (from our laboratory collection) and the yeast-like fungus *Candida albicans* (from our laboratory collection)] on dry surfaces. Taken together, we concluded that the tested bacteria favor lower temperatures for their survival in dry environments. Therefore, the thermal control of dry fomites has the potential to control bacterial survival on high-touch surfaces in hospitals.

**Funding:** This work was supported by the Japan Science and Technology Agency to HY. The funder had no role in study design, data collection and analysis, decision to publish, or preparation of the manuscript.

**Competing interests:** The authors have declared that no competing interests exist.

## Introduction

Strategies to ensure greater control of hospital-acquired infections are urgently needed. Microbe-contaminated dry surfaces in hospitals that are frequently touched by hands, termed 'high-touch' surfaces, such as handrails, doorknobs, lockers, and the table surfaces of inpatients, present a high risk in terms of the spread of nosocomial infections [1–3]. It is well recognized that improved hand hygiene and cleaning of surfaces are critically important in controlling hospital-acquired infections [4–6]. While improved cleaning practices have led to a decrease in the transmission of hospital-acquired infections [7–9], in general, hospital cleanliness is still not sufficient to control infections [10–12]. This is confirmed by the fact that outbreaks of hospital-acquired infections still occur worldwide [13–15]. It is therefore possible that there are other unknown factors responsible for bacterial survival, particularly on dry surfaces, in hospitals.

We previously monitored hospital cleanliness on the surfaces of 752 sites in nurse and patient areas in three hospitals located in the central area of Sapporo, Japan, and revealed the presence of a wide range of organic contamination, including microbial contamination, that could be spread via hand touching [16–18]. Considerable variability in bacterial detection was noted, even for the same surfaces, indicating that the ongoing cleanliness of dry surfaces in hospitals is insufficient [16–18]. Furthermore, accumulating evidence indicates that various human pathogenic bacteria (such as *Pseudomonas*, *Acinetobacter*, *Enterococcus*, *Staphylococcus*, and *Enterobacteriaceae*), including multidrug-resistant bacteria, can survive for prolonged periods on dry surfaces [19–25]. The reason for the variability in bacterial detection remains unclear, but changing physiological factors such as environmental temperature may affect bacterial survival. Whether thermal control can regulate bacterial survival in dry environments remains unknown.

In this study, we monitored the survival of human pathogenic bacteria [*Escherichia coli* ATCC, extended-spectrum β-lactamase (ESBL)-producing *E. coli*, New Delhi metallo-β-lactamase (NDM)-producing *E. coli*, and *Staphylococcus aureus* ATCC] on dry materials (vinyl chloride, aluminum, plastic, stainless steel) over a range of temperatures (room temperature to 37°C). We found that bacteria favor lower temperatures for survival in dry environments. Furthermore, we propose that fomites warmed to human body core temperature may help to control bacterial survival in dry environments, and that such a strategy may prohibit the emergence of human pathogenic bacteria in hospital environments, eventually reducing the need for antibiotics as well as disinfectants.

## Materials and methods

### Bacterial strains

The bacterial strains used in this study were as follows: *E. coli* ATCC 25922, ESBL-producing *E. coli*, NDM-producing *E. coli* [26], and *S. aureus* ATCC 29213. All of the bacteria were maintained on LB agar plates (Nakalai Tesque Inc., Kyoto, Japan) with or without appropriate antibiotics at 37°C. Both the antimicrobial-resistant *E. coli* strains were isolated at Hokkaido University Hospital. The reference strains were purchased from the American Type Culture Collection (ATCC, Manassas, VA, USA). *E. coli* DH5α carrying GFP-expressing plasmid (pBBR122 encoding *gfp*) was also used for this study [27]. *E. coli* DH5α was maintained on LB agar with chloramphenicol (10 μg per ml) to maintain the plasmid. *Pseudomonas aeruginosa* ATCC29213 and *Enterococcus faecalis* ATCC29212 were also purchased from the ATCC. *Acinetobacter baumannii* and *Serattia marcescens* were isolated from Hokkaido University Hospital. *Bacillus subtilis* and the yeast-like fungus *Candida albicans* from our laboratory collection

were also used for this study. All bacterial strains (except for *E. faecalis*) and the yeast-like fungus were maintained on LB agar and PDA agar, respectively. Because of requiring more nutrients, *E. faecalis* was maintained on BHI agar.

## Assessment of bacterial dynamics on vinyl chloride

*S. aureus* ATCC 29213 and *E. coli* ATCC 25922 were separately adjusted in PBS at approximately $10^7$ CFU per ml, and were either mixed or maintained separately in LB broth. Then, 20 μl of each solution ($5.0 \times 10^5$ CFU/spot) was spotted onto vinyl chloride material ($3 \times 3$ cm) that is used commercially for flooring and roofing (thanks to TOLI Corporation, Osaka, Japan and TAJIMA ROOFING, Tokyo, Japan). The bacterial solution was left to dry and then incubated at room temperature (~20˚C) or in an incubator (30˚C and 37˚C) for up to 11 days. In some experiments, the bacterial suspensions were dried onto the material with cardiolipin (1 or 100 μM). After incubation, the dried spots were wiped with cotton swabs, and suspended in PBS. The bacteria were enumerated by spreading onto Mannitol salt agar (for culturing of *S. aureus*) and MacConkey agar (for culturing of *E. coli*), and bacterial numbers were expressed as CFUs per spot. The spots of bacteria on the dry material were also visualized by SEM observations (see below). The amounts of ATP in each spot were monitored by a Clean-Trace Luminometer 3M (Saint Paul, MN, USA), and the values were expressed as bioluminescence relative light units (RLUs), according to a previously described protocol [17].

## Assessment of bacterial dynamics on the other dry fomites (aluminum, plastic, stainless steel)

The survival of *S. aureus* ATCC 29213 and *E. coli* ATCC 25922 was also tested on other materials ($10 \times 10$ cm; aluminum, plastic, stainless steel) at a range of temperatures (15˚C, 30˚C, 37˚C) for up to 11 days. Bacterial numbers were evaluated as stated above. Some experiments on plastic were adjusted to a constant humidity of 40%–60% by placing a 100-mL beaker filled with water into the incubator. However, unfortunately, we were unable to maintain a uniform humidity in this way. The humidity of the incubator without the beaker was less than 20%. Humidity was measured with a general hygrometer.

## Dynamics of multidrug-resistant *E. coli* on dry fomite (vinyl chloride)

The dynamics of both ESBL- and NDM-producing *E. coli* strains with or without *S. aureus* ATCC 29213 on vinyl chloride material were also assessed. The bacteria were dried and then incubated at distinct temperatures (15˚C, 30˚C, 37˚C) for up to 11 days. Bacterial numbers were evaluated as stated above.

## Scanning electron microscopic (SEM) analysis

According to a previously reported method [17], the vinyl chloride material ($1 \times 1$ cm) spotted with bacteria was washed with saline, fixed with 2.5% (v/v) glutaraldehyde in phosphate-buffered saline (pH 7.4) for 2 h at room temperature, then treated with 2% (w/v) osmium tetroxide for 1 h at 4˚C. The samples were then dehydrated in ethanol, freeze-dried, and coated with osmium using a plasma osmium coater, for analysis using a scanning electron microscope (Hitachi S-4800; Hitachi, Tokyo, Japan).

## Bacterial dynamics on the heated handrail device

A stainless steel handrail pipe (diameter, 13 mm; thickness, 2 mm) equipped with a heater wire in the center of pipe to control the dry surface at body core temperature was constructed

using an acrylic-processing assembly kit from FabCloud (Tokyo, Japan). Bacterial dynamics on this surface were assessed as follows. In brief, the GFP-expressing *E. coli* DH5α were spotted at regular intervals (1 cm) onto the handrail device with or without the heater at approximately $10^6$ CFU per spot (10 μl per spot). The handrail device (with or without heat control) was then incubated at room temperature for one day. The dried spots were wiped with cotton swabs, and suspended into PBS. The number of bacteria was evaluated by the GFP signals after spotting onto LB or MacConkey agar plates containing chloramphenicol, and was expressed as CFUs per spot. In addition, the surface temperature on the handrail pipe was monitored using a hand-held infrared sensor (CT-2000D; Custom Co., Ltd., Tokyo, Japan) and a high-resolution infrared sensor (InfReC R300; NEC Avio Infrared Technologies Co., Ltd., Tokyo, Japan) installed with appropriate software (InfReC Analyzer NS9500; NEC Avio Infrared Technologies Co., Ltd.). We also assessed whether the heated handrail device could work for the illumination of the other bacteria (*S. aureus*, *P. aeruginosa*, *E. faecalis*, *A. baumannii*, *S. marcescens* and *B. subtilis*) and the yeast-like fungus (*C. albicans*). As mentioned above, the survival of bacteria (*P. aeruginosa*, *A. baumannii*, and *B. subtilis*) and the yeast-like fungus after drying on the device was monitored on LB agar and PDA agar, respectively. The survival of *S. aureus* and *S. marcescens* was also monitored on Mannitol salt agar and MacConkey agar, respectively. Meanwhile, because of requiring more nutrients, *E. faecalis* was monitored on BHI agar.

## Statistical analysis

Multiple comparisons of the data were assessed by Bonferroni/Dunn analysis. A *p* value of <0.05 was considered statistically significant. All calculations were conducted using Excel for Mac (2011) with Statcel3C.

## Results

### Representative human pathogenic bacteria favor lower temperatures for their survival under dry conditions

To assess the survival of representative human pathogens under dry conditions, we monitored changes in bacterial numbers [*E. coli* (ATCC 25922), *S. aureus* (ATCC 29213), or a mixture of these bacteria (because it is possible that they are found together on 'high-touch' surfaces)] on vinyl chloride, a widely used material in everyday items, at distinct temperatures [room temperature (around 20°C), 30°C, 37°C] for 11 days. Bacterial numbers were assessed using the following indicators: the number of colony-forming units (CFUs), the amount of ATP, and morphological changes (as determined by SEM analysis) (Fig 1A). Immediately (within 20 min) after drying, the CFU numbers of *E. coli*, but not *S. aureus*, showed a two-log reduction; however, both bacteria, either singularly or in a mixture, survived and retained the potential to be spread (Fig 1B). The number of CFUs decreased dramatically over time for both bacteria on dry surfaces, but this trend was amplified by an increase in temperature (Fig 1C). Interestingly, the presence of *S. aureus* appeared to enhance the survival of *E. coli* in the bacterial mixture compared with *E. coli* alone (Fig 1C, See '†'). SEM observations revealed that the morphological changes in both bacteria, singularly or in a mixture, one day after drying was minimal, supporting the CFU assay results and indicating survival of both bacteria (Fig 2). ATP assessment revealed no significant differences for either bacteria with varying temperature (S1 Fig). Also, cardiolipin, an agent that aids the survival of *S. aureus* in dry environments [28, 29], did not support the survival of *E. coli* after drying, suggesting the presence of another mechanism independent of cardiolipin (S2 Fig). In addition, although the trend observed with altered temperature did not change, humidity control (40–60%) had the opposite effect on the survival of

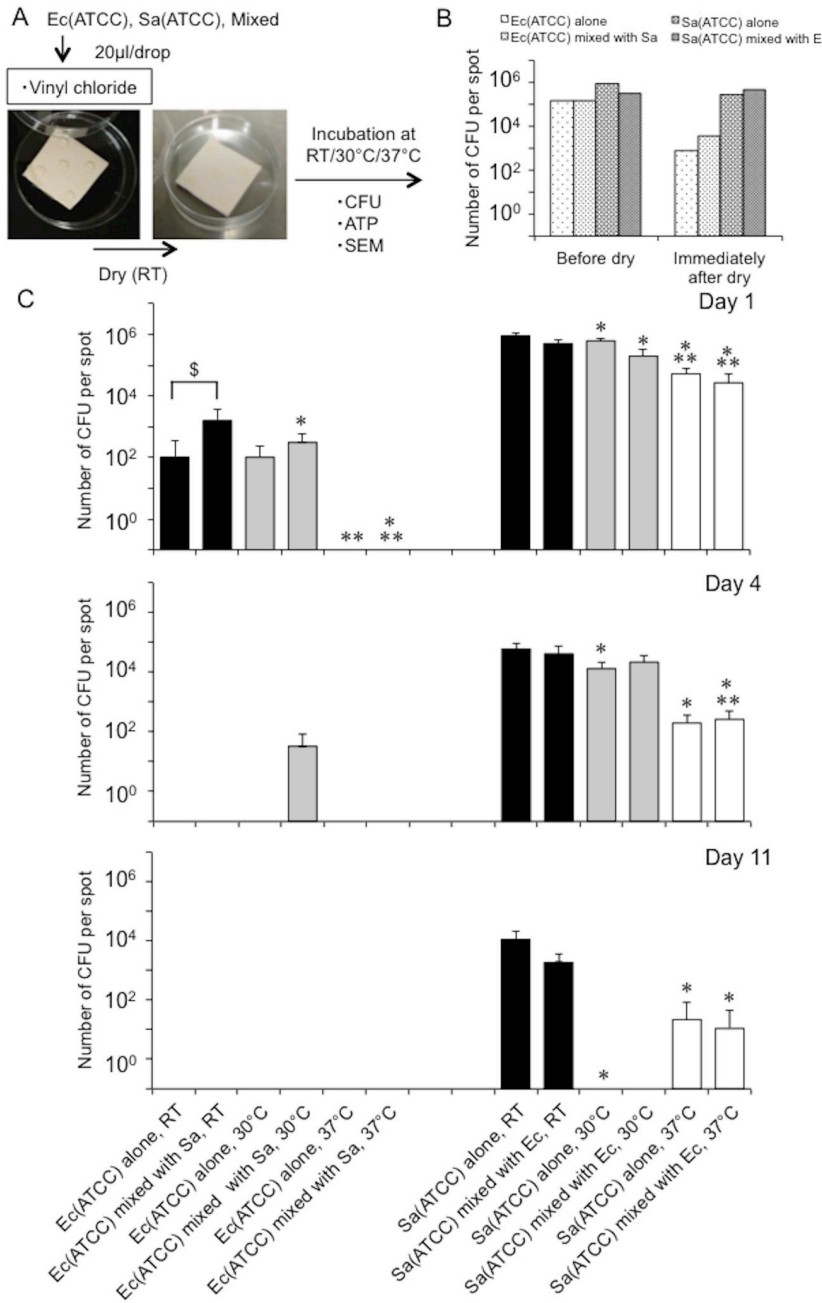

**Fig 1. Representative human pathogenic bacteria favor lower temperatures for their survival on vinyl chloride surfaces. A**. Experimental protocol. Bacterial survival was monitored for mixed or single bacteria on vinyl chloride at distinct temperatures [RT (around 20˚C), 30˚C, 37˚C] for 11 days by a CFU assay (CFU), ATP assessment (ATP) and morphological changes (SEM). Ec, *E. coli* (ATCC 25922). Sa, *S. aureus* (ATCC 29213). **B**. Bacterial numbers immediately (within 20 min) after drying. Ec, *E. coli* (ATCC 25922). Sa, *S. aureus* (ATCC 29213). **C**. Changes in bacterial numbers at distinct temperatures [RT (around 20˚C), 30˚C, 37˚C] over 11 days, as determined by a CFU assay. Ec, *E. coli* (ATCC 25922). Sa, *S. aureus* (ATCC 29213). *, $p < 0.05$ vs. the values at room temperature. **, $p < 0.05$ vs. 30˚C. †, $p < 0.05$ between the bars.

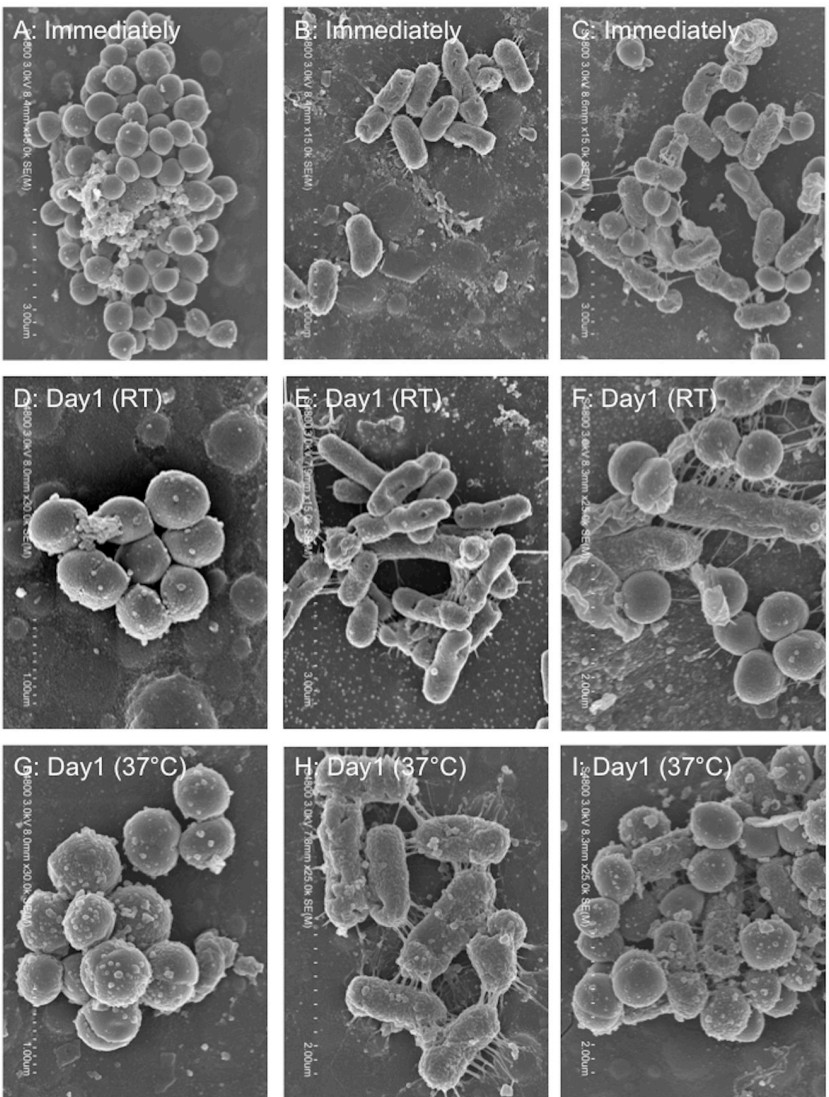

**Fig 2. Representative SEM images showing bacterial morphologies on vinyl chloride surfaces.** *E. coli* alone (A, D, G). *S. aureus* alone (B, E, H). Mixture of *E. coli* with *S. aureus* (C, F, I). Scale bars in panels, 1 μm (D, G), 2 μm (F, H, I) and 3 μm (A, B, C, E). RT, room temperature.

microorganisms on dry surfaces (S3 Fig). Specifically, while the humidity control weakened the thermal effect on the survival of *S. aureus* on dry environments, a synergistic effect of inhibition on the survival of *E. coli* was observed. Thus, our findings showed that these bacteria favor a lower temperature for their survival in dry environments, and humidity has a lesser effect.

## Differences in the surface material had no effect on bacterial survival (including multidrug-resistant bacteria) in dry environments over a range of temperatures

Next, to confirm whether the effect of temperature on bacterial survival on dry surfaces is universal regardless of the material surface, we monitored changes in bacterial numbers [*E. coli*

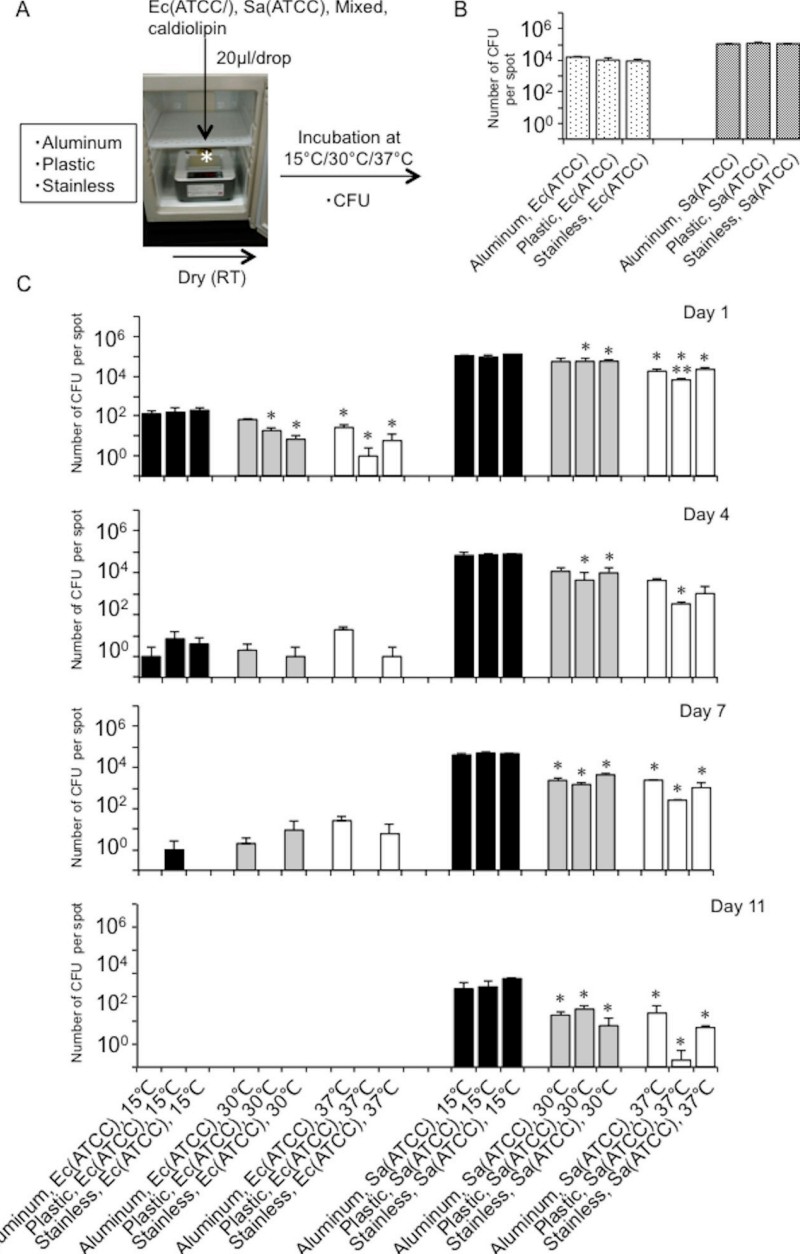

**Fig 3. Representative human pathogenic bacteria favor lower temperatures for their survival on other surfaces (aluminum, plastic, stainless steel). A**. Experimental protocol. Bacterial survival was monitored for mixed or single bacteria on other surfaces (aluminum, plastic, stainless steel) at distinct temperature (15°C, 30°C, 37°C) for 11 days by a CFU assay (CFU). Ec, *E. coli* (ATCC 25922). Sa, *S. aureus* (ATCC 29213). **B**. Bacterial numbers immediately (within 20 min) after drying. Ec, *E. coli* (ATCC 25922). Sa, *S. aureus* (ATCC 29213). **C**. Changes in bacterial numbers at distinct temperatures (15°C, 30°C, 37°C) over 11 days, as determined by a CFU assay. Ec, *E. coli* (ATCC 25922). Sa, *S. aureus* (ATCC 29213). *, $p < 0.05$ vs. the values at room temperature. **, $p < 0.05$ vs. 30°C.

(ATCC 25922), *S. aureus* (ATCC 29213), or a mixture] on a range of common materials (vinyl chloride, aluminum, plastic, stainless steel) at distinct temperatures (15°C, 30°C, 37°C) for 11 days, by assessing the CFU numbers (Fig 3A). Immediately (approximately 1 h) after drying, no differences in bacterial numbers were detected between the different materials (Fig 3B).

Although the CFU numbers of both bacteria dramatically reduced over subsequent days on the dry surfaces, this temperature-dependent trend was unaffected by the type of material surface (Fig 3C). Thus, the material itself (vinyl chloride, aluminum, plastic, stainless steel) had no effect on the diminishing bacterial numbers under dry conditions. We also assessed whether the same trend was seen for multidrug-resistant human pathogenic bacteria using ESBL- and NDM-producing *E. coli* isolated from Hokkaido University Hospital. The CFU assay confirmed increased survival of multidrug-resistant bacteria at lower temperature, as well as increased survival in the presence of *S. aureus* (Fig 4). Taken together, we conclude that bacteria favoring a lower temperature for their survival in dry environments is a universal phenomenon, which is unaffected by the surface material or bacterial type. Importantly, our data highlighted the potential for controlling the survival of bacteria on high-touch dry surfaces by warming dry fomites to human body core temperature.

### Handrail device warmed to human body core temperature prohibited the prolonged survival of *E. coli* on this dry surface

To test this possibility, we constructed a stainless steel handrail pipe (diameter, 13 mm; thickness, 2 mm) equipped with a heater wire in the center of the pipe to control the dry surface to body core temperature (S4A Fig). Assessment with a hand-held infrared sensor showed that the surface temperature on the handrail pipe was controlled to almost body core temperature, although the temperature on the device surface varied from the end to the center of the pipe (range: 30°C–40°C) (S4B Fig). A heatmap generated with the high-resolution infrared sensor showed that in contrast to the stainless steel pipe without the heater, the surface temperature on the heated pipe was maintained at body core temperature (Fig 5A), indicating that it is an effective tool to monitor the influence of temperature on bacterial survival on a dry surface. We monitored the survival of GFP-expressing *E. coli* (DH5α) spotted onto the handrail surface at regular intervals (1 cm) compared with the GFP signals at equivalent spots on an LB agar plate, and assessed CFU numbers. GFP signals were dramatically decreased on the heated handrail compared with the unheated handrail even 6 h after drying, and the signals were virtually undetectable one day after drying (Fig 5B). Consistent with this, the CFU numbers on the heated handrail were significantly reduced compared with those on the unheated handrail (Fig 5C). In addition, we assessed whether the warming handrail could work on other bacteria. The results indicated that the prolonged survival of bacteria (*E. faecalis*, *S. aureus*, *P. aeruginosa*, *A. baumannii*, and *S. marcescens*) as well as *E. coli*, was significantly prohibited, although no effective elimination of the spore-forming bacterium (*B. subtilis*) was detected (Fig 6). We also confirmed that the yeast-like fungus (*C. albicans*) failed to survive immediately (within 6 h) after drying on the device (S5 Fig). Taken together, we concluded that the handrail device warmed to human body core temperature prohibited the prolonged survival of some bacteria and a yeast-like fungus responsible for nosocomial infections on a dry surface.

## Discussion

Currently, in the US, an estimated 2 million hospital-acquired infections are reported annually, associated with $5 billion of additional healthcare costs. The control of nosocomial infections is therefore a priority for healthcare providers worldwide [30]. Here, we showed, for the first time, that a lower temperature is favored for the survival of both *E. coli* and *S. aureus* on dry fomites. We also demonstrated that a handrail device warmed to human body core temperature could prohibit the prolonged survival of several pathogens responsible for nosocomial infection (*E. coli*, *S. aureus*, *E. faecalis*, *P. aeruginosa*, *A. baumannii* and *S. marcescens*) and a yeast-like fungus (*C. albicans*) on this dry surface. Innovations such as the development of

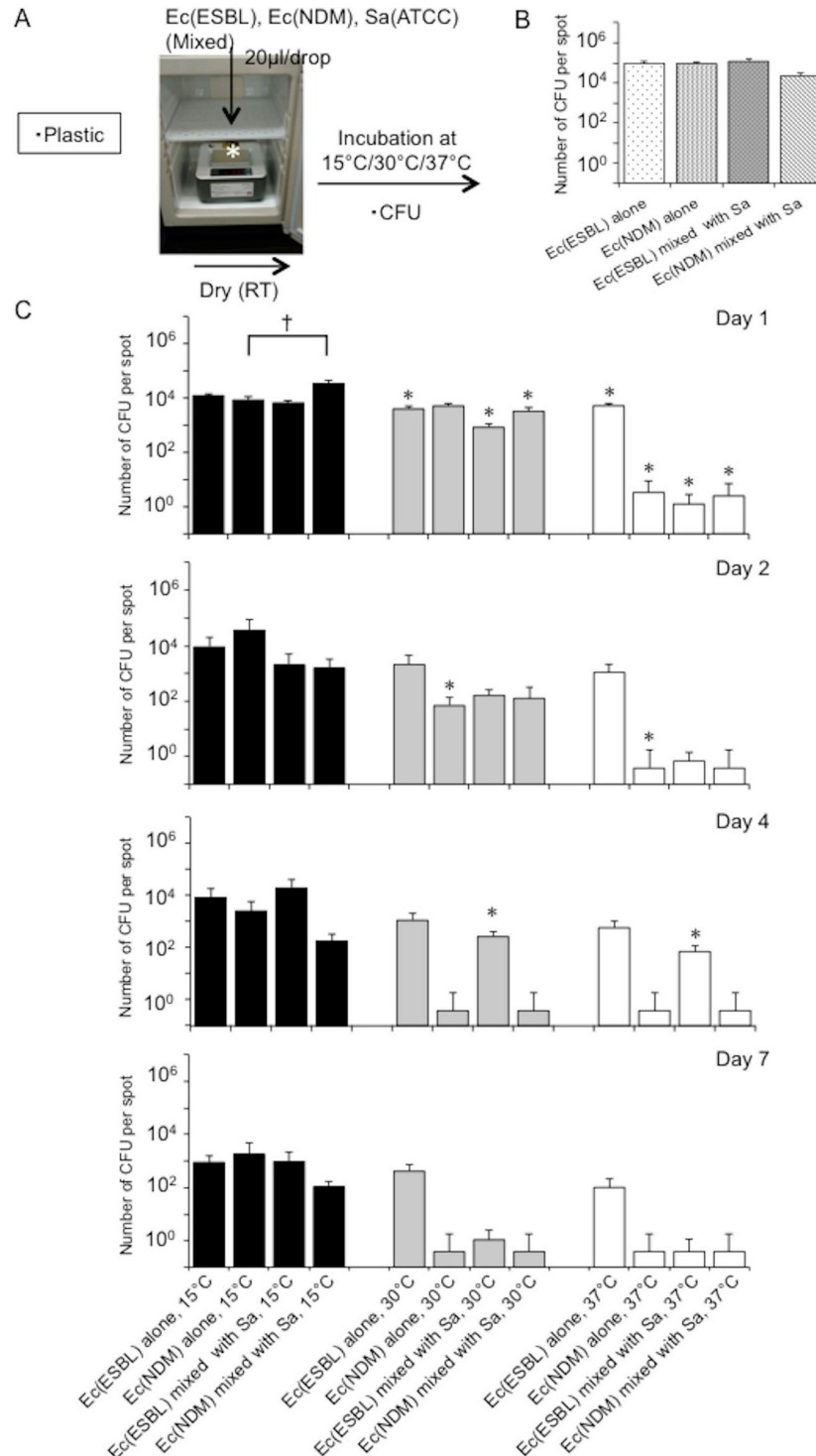

**Fig 4. Representative multidrug-resistant bacteria also favor lower temperatures for their survival on plastic surfaces. A**. Experimental protocol. Bacterial survival was monitored for mixed or single bacteria on plastic surfaces at distinct temperatures (15˚C, 30˚C, 37˚C) for 11 days by a CFU assay (CFU). Ec (ESBL), ESBL-producing *E. coli*. Ec (NDM), NDM-producing *E. coli*. Sa, *S. aureus* (ATCC 29213). **B**. Bacterial numbers immediately (within 20 min) after drying. Ec (ESBL), ESBL-producing *E. coli*. Ec (NDM), NDM-producing *E. coli*. Sa, *S. aureus* (ATCC 29213). **C**. Changes in bacterial numbers at distinct temperatures (15˚C, 30˚C, 37˚C) over 11 days, as determined by a CFU assay. Ec (ESBL), ESBL-producing *E. coli*. Ec (NDM), NDM-producing *E. coli*. Sa, *S. aureus* (ATCC 29213). *, $p < 0.05$ vs. the values at room temperature. †, $p < 0.05$ between the bars.

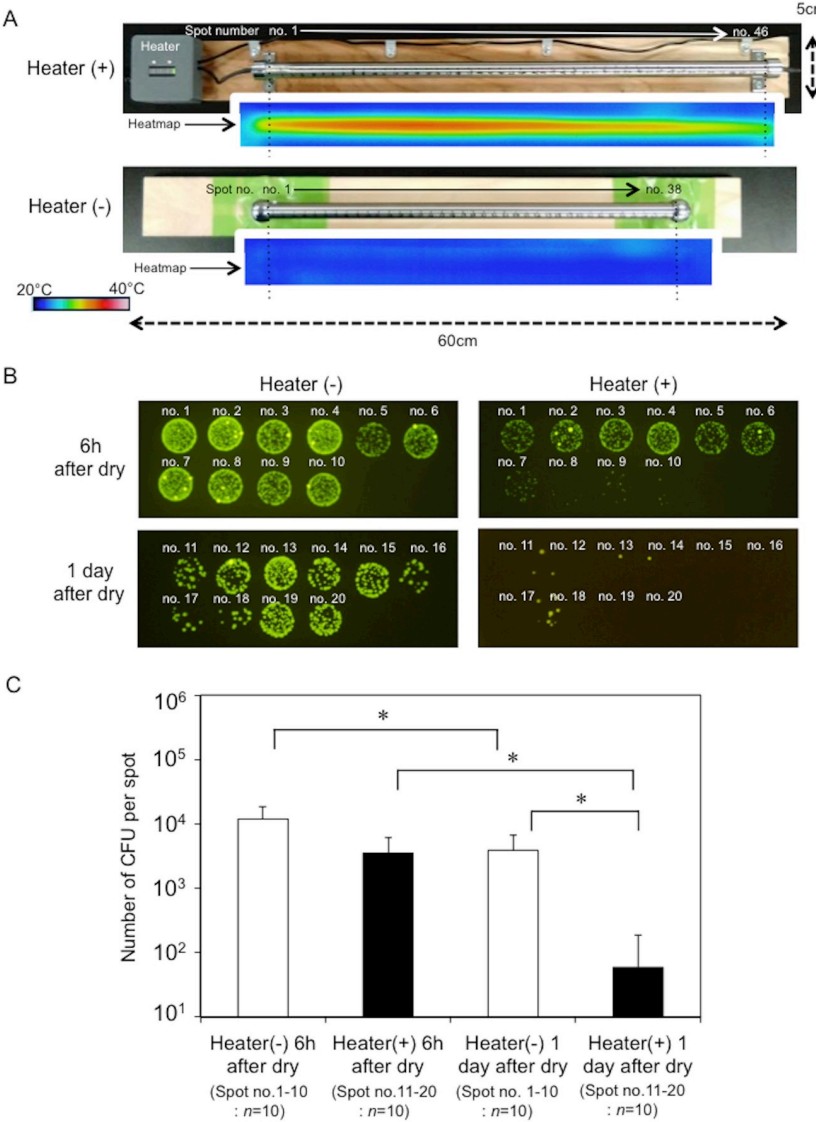

**Fig 5. Handrail device warmed to human body core temperature prohibited the prolonged survival of GFP-expressing *E. coli* on a stainless steel surface. A**. Heatmaps of the handrail device with or without a heater. The heatmaps were constructed using a high-resolution infrared sensor (InfReC R300) installed with appropriate software (InfReC Analyzer NS9500). **B**. Representative images showing decreasing GFP-signals for the bacteria collected from the stainless steel surface warmed by the heater. Numbers show the sampling positions on the pipe. Heater (-), handrail device without heater. Heater (+), handrail device with heater. **C**. Comparison of the bacterial CFU numbers on the stainless steel device with and without the heater. Black and white bars show the CFU values for the device with and without the heater. *, $p < 0.05$ between the bars.

heated fomites, for example, heated handrails or doorknobs, may prevent the survival of bacteria responsible for nosocomial infections on dry hospital environments, without the need for disinfectants.

It is well established that decreasing environmental temperature has a significant impact on the survival of pathogens in soil or water, sometimes leading to bacterial persistence [31–33], but whether temperature influences bacterial survival on artificial materials, such as dry fomites in hospitals, remained unknown. Other factors such as UV radiation, humidity, the presence of organic materials, and surface type are known to be associated with the ability of

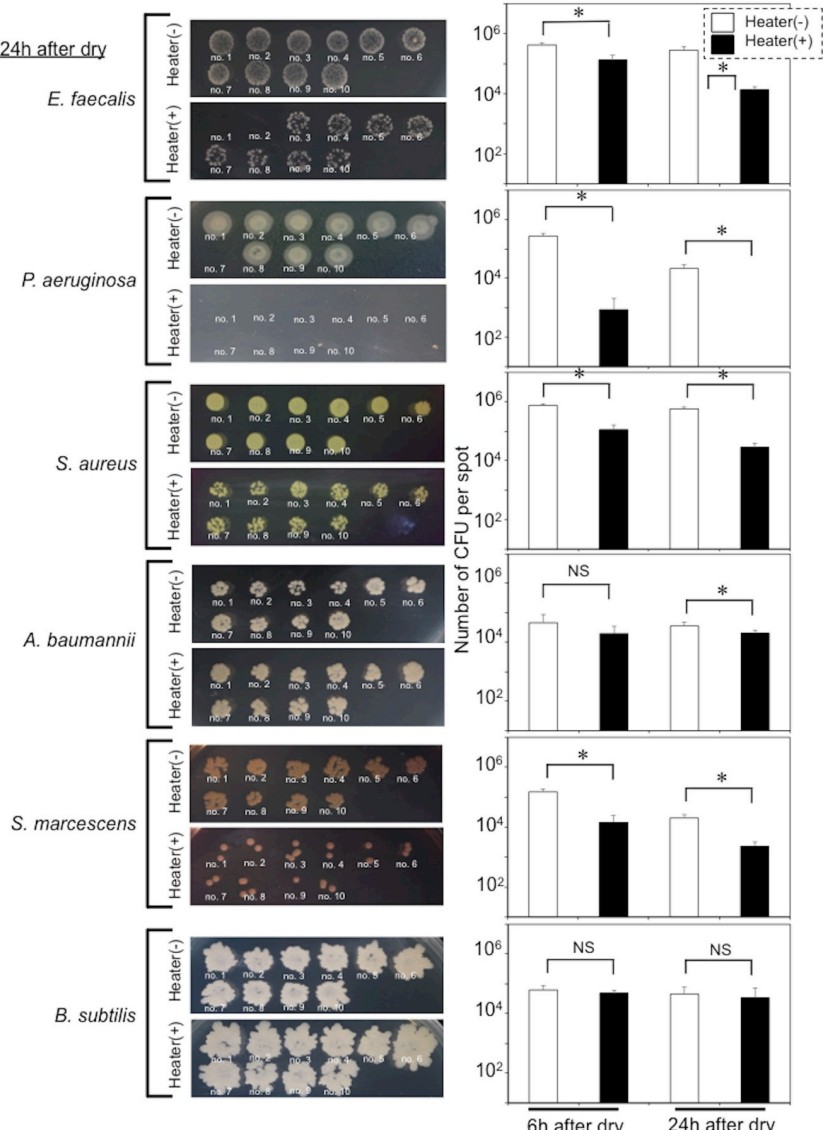

**Fig 6. Handrail device warmed to human body core temperature prohibited the prolonged survival of other bacteria without spore formation on a stainless steel surface.** The other bacteria (*E. faecalis*, *S. aureus*, *P. aeruginosa*, *A. baumannii*, *S. marcescens* and *B. subtilis*) were used for this study (see the Materials and methods). Images obtained at 24 h after drying correspond to representative bacterial images showing the effect of thermal control on survival of each of the bacteria on the handrail device. Numbers (1–10) indicate bacterial spots. Graphs (right) show a comparison of the bacterial CFU numbers on the stainless steel device with [Heater(+)] and without the heater [Heater(-)]. Black and white bars show the CFU values for the device with and without the heater. *, $p < 0.05$ between the bars. NS, no significant difference.

bacteria responsible for hospital-acquired infections to survive on dry surfaces [34–36]; however, temperature is a factor that can be easily and accurately controlled. Furthermore, a temperature of 37˚C, which is the core body temperature, is a comfortable temperature that is not harmful to the touch [37]. For these reasons, we selected temperature as a critical factor to study its effects (in particular, body core temperature) on bacterial dynamics and survival on dry fomites.

The numbers of CFUs of both bacteria (*E. coli* and *S. aureus*), regardless of whether or not they carried multidrug-resistant genes, were dramatically reduced during the days after drying, and this decrease in bacterial numbers was significantly promoted by increased temperature. Because water evaporation from bacterial cells can be accelerated on dry fomites at high temperature, we speculate that it is difficult for bacteria to maintain metabolic activity under such conditions; thus, their viability decreases. We found that Gram-positive *S. aureus* could survive longer than Gram-negative *E. coli*, potentially indicating that cell wall thickness plays a role in prolonging the survival of *S. aureus* on dry fomites. At lower temperature, *E. coli* survived for a few days whereas *S. aureus* survived for over 10 days. Although the survival period for these bacteria is limited, particularly for *E. coli*, it is still sufficient time for these bacteria to transfer from person to person via high-touch surfaces.

Because of its ease of monitoring, ATP bioluminescence has been employed as a visual indicator of general organic contamination of high-touch surfaces in hospitals [38–42]. However, our assessment of ATP showed that, in contrast to the CFU assay, no significant differences in ATP were associated with changes in temperature. Evidence from our previous studies suggests that ATP may be physically stable at a range of temperatures [16, 17]. Furthermore, ATP derived not only from bacteria but also from other sources such as fungi or dead cells may accumulate on high-touch surfaces. Taken together, we concluded that ATP is an unreliable measure for monitoring pathogenic bacteria on high-touch surfaces in hospitals.

Interestingly, the presence of *S. aureus* in the bacterial mixture promoted the survival of *E. coli* compared with *E. coli* alone. By contrast, cardiolipin, which is an effective factor promoting *S. aureus* survival in dry environments [28, 29], had no effect on supporting the survival of *E. coli* after drying, suggesting the presence of another mechanism independent of cardiolipin. Although this mechanism remains unknown, coexistence with cocci, which have a smaller surface area than bacilli and therefore retain water more effectively, may aid the survival of bacilli on dry surfaces. High-touch dry surfaces are invariably contaminated with *S. aureus* because this organism is a ubiquitous inhabitant of the human body, and it is therefore crucial to understand its dynamics on dry surfaces. Cardiolipin is a membrane component that enhances stress tolerance, thereby maintaining bacterial viability [43–45], and several studies have shown that cardiolipin is involved in biofilm formation [46, 47]. We therefore assessed whether the presence of cardiolipin affected the survival of *E. coli* on dry surfaces, but no impact was detected indicating that other factors may be involved.

Although bacterial numbers dramatically reduced each day on dry surfaces, the degree of this reduction varied slightly between experiments and the degree to which *S. aureus* promoted the survival of *E. coli* also varied between experiments. This indicated that there may be factors other than temperature that regulate bacterial survival on dry fomites. Humidity, which can block water evaporation, has also been reported to have an impact on bacterial survival [34–36, 48, 49]. Our experiments showed that constant humidity (40%–60%) affected the survival of bacteria on dry surfaces. Whereas, after prolonged incubation without humidity control (less than 20% humidity) (See Fig 1C), variation in bacterial numbers appeared to be increased, suggesting that low humidity can favor the survival of small numbers of bacteria on dry materials. However, because of the difficulties in controlling humidity, we did not accurately adjust humidity in our experiments. Therefore, further studies are needed in which humidity, as well as temperature, are more precisely controlled.

As expected, we confirmed that the warming handrail adjusted to body core temperature prohibited the prolonged survival of several nosocomial-related pathogens including *E. coli*, *S. aureus*, *E. faecalis*, *P. aeruginosa*, *A. baumannii*, *S. marcescens* and *C. albicans*. The heated handrail device used in this study was constructed using an acrylic board processing equipment kit (see Materials and methods), which cost 5,680 yen (JPY) (approximately $60).

Because this device was cheap and simple to construct, involving a stainless steel handrail pipe (diameter, 13 mm; thickness, 2 mm) equipped with a heater wire in the center of the pipe to control the dry surface at body core temperature, we believe that this technology may be applicable for the development of heated handrails for use in hospitals to control hospital-acquired infections. One limitation of this device was that the temperature varied from the end to the center of the pipe (range: 30°C-40°C), meaning that the temperature on the device surface was not constant, as indicated by the differences in bacterial numbers spotted onto the agar plates from each of the samples collected from the handrail. Further improvements in the design of this device are therefore needed prior to practical use.

In conclusion, we demonstrated that human pathogens (*E. coli*, *S. aureus*, *E. faecalis*, *P. aeruginosa*, *A. baumannii*, *S. marcescens* and *C. albicans*), responsible for nosocomial infections. favor a lower temperature for their survival in dry environments. Based on this finding, we proposed a novel and simple strategy involving the warming of high-touch fomites in hospitals to body core temperature, to control the survival of human pathogenic bacteria responsible for nosocomial infections. However, this study had several limitations. As mentioned above, humidity, which can also influence bacterial survival, was not accurately controlled throughout our experiments [34–36, 48, 49], and also the range of human pathogens used for this study were limited. Further studies are needed to clarify the effect of humidity on bacterial survival under dry conditions and to confirm the reproducibility of our findings in other bacteria. Further innovations are also needed to develop handrails or doorknobs with a uniform surface temperature equivalent to that of the core human body temperature before practical application in hospital environments.

## Supporting information

**S1 Fig. ATP assessment of the bacterial spots on the vinyl chloride material showed that there were no significant differences corresponding to incubation times or temperature.** The amount of ATP was determined using a Clean-Trace Luminometer (3M, USA), and the values were expressed as relative light units (RLUs), according to a previously described protocol [17]. Ec, *E. coli* ATCC 25922. Sa, *S. aureus* ATCC 29213. RT, room temperature.
(PDF)

**S2 Fig. Cardiolipin had no effect on *E. coli* survival after drying.** Black and dashed bars show the CFU numbers of *E. coli* and *S. aureus*, respectively. **A**. Bacterial numbers immediately (within 20 min) after drying. Ec, *E. coli* (ATCC 25922). Sa, *S. aureus* (ATCC 29213). **B**. Changes in bacterial numbers at distinct temperatures (15°C, 30°C, 37°C) over 11 days incubation, as analyzed by a CFU assay. Ec, *E. coli* (ATCC 25922). Multiple comparisons of the data were assessed by Bonferroni/Dunn analysis. Asterisk indicates a statistically significant difference compared with the value at 15°C. *, $p<0.05$. There was no statistically significant difference between the values at 30°C, regardless of the presence of cardiolipin. †, $p<0.05$. Ec, *E. coli* ATCC 25922. Sa, *S. aureus* ATCC 29213.
(PDF)

**S3 Fig. Effect of humidity on the survival of bacteria (*E. coli* and *S. aureus*) on dry environments (plastic).** Changes in bacterial numbers at distinct temperatures (15°C, 30°C, 37°C) with or without humidity control over 7 days, as determined by a CFU assay. **Upper panel**. Ec, *E. coli* (ATCC 25922). **Lower panel**. Sa, *S. aureus* (ATCC 29213). *, $p<0.05$ vs. the values without humidity control. See the Materials and methods.
(PDF)

**S4 Fig. Surface temperature on the handrail pipe with or without the heater. A**. Structure of the handrail pipe with the heater. The stainless steel handrail pipe (diameter, 13 mm; thickness, 2 mm) was equipped with a heater wire at the center of the pipe to control the dry surface at body core temperature. **B**. Comparison of the surface temperature of the handrails with and without heaters. Temperatures were monitored by a hand-held infrared sensor (CT-2000D). Each of the values is shown as an average with the standard deviation. Black and white circles are shown as the values with and without heaters, respectively.
(PDF)

**S5 Fig. Handrail device warmed to human body core temperature prohibited the prolonged survival of yeast-like fungus (*C. albicans*).** Images show the effect of thermal control on the survival of *C. albicans* on the handrail device 6 and 24 h after drying with [Heater (+)] or without thermal control [Heater (−)]. Numbers (no. 1–10) show the yeast-like fungus spots on the PDA agar.
(PDF)

## Acknowledgments

We thank the Edanz Group (www.edanzediting.com/ac) for editing a draft of this manuscript.

## Author Contributions

**Conceptualization:** Tomoko Shimoda, Torahiko Okubo, Rika Yano, Jeewan Thapa, Hiroyuki Yamaguchi.

**Data curation:** Tomoko Shimoda, Torahiko Okubo, Rika Yano, Hiroyuki Yamaguchi.

**Formal analysis:** Yoshiki Enoeda, Hiroyuki Yamaguchi.

**Funding acquisition:** Hiroyuki Yamaguchi.

**Investigation:** Tomoko Shimoda, Torahiko Okubo, Yoshiki Enoeda, Shinji Nakamura, Hiroyuki Yamaguchi.

**Methodology:** Tomoko Shimoda, Torahiko Okubo, Shinji Nakamura.

**Supervision:** Hiroyuki Yamaguchi.

**Validation:** Torahiko Okubo, Hiroyuki Yamaguchi.

**Visualization:** Shinji Nakamura.

**Writing – original draft:** Hiroyuki Yamaguchi.

**Writing – review & editing:** Jeewan Thapa.

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
