## [Decision Letter · Decision Letter 0]

4 Oct 2019

PONE-D-19-18407

Effect of Thermal Control of Dry Fomites on Regulating the Survival of Human Pathogenic Bacteria Responsible for Nosocomial Infections

PLOS ONE

Dear Dr. Yamaguchi,

Thank you for submitting your manuscript to PLOS ONE. After careful consideration, we feel that it has merit but does not fully meet PLOS ONE’s publication criteria as it currently stands. Therefore, we invite you to submit a revised version of the manuscript that addresses the points raised during the review process.

Although the manuscript makes important observations regarding survival of bacterial pathogens on a variety of surfaces at different temperatures, there are concerns about the limited scope of the pathogens studied, and over interpretation of the findings as indicated in the comments from Reviewer 1. Thus, if you choose to resubmit a revised manuscript this should include analysis of additional pathogens as suggested by Reviewer 1.  

In Figure 1, Day 11, bacteria are surviving at 37 degrees C, but not at 30 degress C. Please provide a potential explanation for this why this difference is occurring since the data from the other reported experiments suggest less survival at 37 degrees C.

It is assumed that the temperature of 37 degrees C was chosen to heat fomites to because this would be a temperature that would be comfortable to human touch. The temperature of skin on the hand is approximately 30 degrees C, 37 degrees C is human body core temperature. The question arises as to what is the subjective sensation to touch of an object at 37 degrees C. Please cite and include information from touch temperature studies which measure the subjective and objective effects of temperature on human skin.

Humidity may be a factor in bacterial survival on fomites as acknowledged in the manuscript.  Although this variable may be difficult to control in carrying out some of the experiments, the ambient humidity should be easily measured when carrying out the experiments involving bacterial survival on the simulated, heated handrail. This needs to be recorded at the time of these experiments and reported in the manuscript.

The esthetic appearance of the graphs needs to be improved, particularly be modifying to provide a more streamlined numerical representation of y axis intervals.  

We would appreciate receiving your revised manuscript by November 15, 2019. To enhance the reproducibility of your results, we recommend that if applicable you deposit your laboratory protocols in protocols.io, where a protocol can be assigned its own identifier (DOI) such that it can be cited independently in the future. For instructions see: http://journals.plos.org/plosone/s/submission-guidelines#loc-laboratory-protocols

We look forward to receiving your revised manuscript.

Kind regards,

Thomas Byrd

Academic Editor

PLOS ONE

**Journal Requirements:**

2. Our editorial staff has assessed your submission, and we have concerns about the grammar, usage, and overall readability of the manuscript.  We therefore request that you revise the text to fix the grammatical errors and improve the overall readability of the text before we send it for review. We suggest you have a fluent, preferably native, English-language speaker thoroughly copyedit your manuscript for language usage, spelling, and grammar.

If you do not know anyone who can do this, you may wish to consider employing a professional scientific editing service.  

Whilst you may use any professional scientific editing service of your choice, PLOS has partnered with both American Journal Experts (AJE) and Editage to provide discounted services to PLOS authors. Both organizations have experience helping authors meet PLOS guidelines and can provide language editing, translation, manuscript formatting, and figure formatting to ensure your manuscript meets our submission guidelines. To take advantage of our partnership with AJE, visit the AJE website (http://learn.aje.com/plos/) and enter referral code PLOS15 for a 15% discount off AJE services. To take advantage of our partnership with Editage, visit the Editage website (www.editage.com) and enter referral code PLOSEDIT for a 15% discount off Editage services. If the PLOS editorial team finds any language issues in text that either AJE or Editage has edited, the service provider will re-edit the text for free.

Please note that PLOS ONE does not copyedit accepted manuscripts and that one of our criteria for publication is that articles must be presented in an intelligible fashion and written in clear, correct, and unambiguous English (http://www.plosone.org/static/publication#language). If the language is not sufficiently improved, we may have no choice but to reject the manuscript without review.

**Comments to the Author**

1. Is the manuscript technically sound, and do the data support the conclusions?

Reviewer #1: Partly

2. Has the statistical analysis been performed appropriately and rigorously? 

Reviewer #1: I Don't Know

3. Have the authors made all data underlying the findings in their manuscript fully available?

Reviewer #1: Yes

4. Is the manuscript presented in an intelligible fashion and written in standard English?

Reviewer #1: Yes

5. Review Comments to the Author

Reviewer #1: This manuscript addresses the effect of different temperatures on the survival on dry materials of 2 species of bacteria associated with nosocomial infections.

However, some issues should be considered:

Lines 56-58 and 91-93: Even though E. coli and S. aureus are the most frequent nosocomial infection isolates, they do not represent the majority of cases. As such, microorganismos such as Enterococcus spp, P. aeruginosa, A. baumannii, Klebsiella spp, coag neg Staphilococci, Candida spp and C. difficile could be included in order to have representativeness of other Gram positive cocci, Enterobacteriaceae, non-Enterobacteriaceae Gram negative bacilli, yeasts and anaerobes. Therefore, claiming that the "thermal control of dry fomites has the potential to control bacterial survival on high-touch surfaces in hospitals, therefore reducing the spread of nosocomial infections" may sound abusive.

Line 99: Please consider materials and methods before results.

Lines 123-124: Results of the study demonstrate that some strains of S aureus and E coli favor a lower temperature for surviving on dry surfaces... please consider that the spread of nosocomial infection was not addressed.

Lines 253-255: Again, the generalisation that "warming of fomites in hospitals to skin temperature, to control the survival of human pathogenic bacteria responsible for nosocomial infections" sounds excessive.

6. PLOS authors have the option to publish the peer review history of their article (what does this mean?). If published, this will include your full peer review and any attached files.

Reviewer #1: No

---

## [Author Response · Author response to Decision Letter 0]

18 Nov 2019

RESPONSE TO COMMENTS

PONE-D-19-18407 entitled: Effect of Thermal Control of Dry Fomites on Regulating the Survival of Human Pathogenic Bacteria Responsible for Nosocomial Infections

REPLIES TO ACADEMIC EDITOR’S COMMENTS

Comment 1:

“Although the manuscript makes important observations regarding survival of bacterial pathogens on a variety of surfaces at different temperatures, there are concerns about the limited scope of the pathogens studied, and over interpretation of the findings as indicated in the comments from Reviewer 1. Thus, if you choose to resubmit a revised manuscript this should include analysis of additional pathogens as suggested by Reviewer 1.”

As mentioned in the response to the reviewer 1 (See below), and as advised by the editor’s comment, we performed additional experiments with several bacteria (E. faecalis, S. aureus, P. aeruginosa, A. baumannii, S. marcescens and B. subtilis) and a yeast-like fungus (C. albicans). Specifically, the survival of these human pathogens on handrail device was monitored. As a result, the prolonged survival of bacteria (E. faecalis, S. aureus, P. aeruginosa, A. baumannii, and S. marcescens) as well as E. coli, were significantly inhibited, although no effective elimination against spore-forming bacteria (B. subtilis) was seen (Fig 6). In addition, yeast-like fungus, C. albicans, failed to survive immediately after it was added on the device (S5 Fig). We have reflected these results in the revised manuscript with a figure (Fig. 6) and supporting information (S5 Fig). 

Comment 2:

“In Figure 1, Day 11, bacteria are surviving at 37 degrees C, but not at 30 degrees C. Please provide a potential explanation for this why this difference is occurring since the data from the other reported experiments suggest less survival at 37 degrees C.”

In the original work, incubator humidity was not controlled. In the revision, we measured the humidity in the humidity-uncontrolled incubator used for this study. As a result, the humidity of the incubator was found to be less than 20%. As pointed out by the reviewer, when left for a long time (See Fig. 1C), the variation of remaining the bacterial numbers appeared to be slightly increased or decreased irrespective of temperature. We therefore speculate that humidity can influence the presence of small amounts of bacteria on dry materials, specifically implying that low humidity can favor the survival of small numbers of bacteria on dry materials. We have explained it in the revised manuscript.

Comment 3:

“It is assumed that the temperature of 37 degrees C was chosen to heat fomites to because this would be a temperature that would be comfortable to human touch. The temperature of skin on the hand is approximately 30 degrees C, 37 degrees C is human body core temperature. The question arises as to what is the subjective sensation to touch of an object at 37 degrees C. Please cite and include information from touch temperature studies which measure the subjective and objective effects of temperature on human skin.”

As reported previously [37], the temperature of 37°C, which is the human body core temperature, can be felt comfortably regardless of the surrounding temperature when a human touches an object with hands. We believe that this temperature will be the optimum temperature when warming an object touched by a human. We have explained it and have cited the reference in the revised manuscript. Also, according to the comment, we have changed the description of “skin temperature” to “body core temperature” in the revised manuscript.

Comment 4:

“Humidity may be a factor in bacterial survival on fomites as acknowledged in the manuscript. Although this variable may be difficult to control in carrying out some of the experiments, the ambient humidity should be easily measured when carrying out the experiments involving bacterial survival on the simulated, heated handrail. This needs to be recorded at the time of these experiments and reported in the manuscript.”

According to the comment, we have assessed if humidity is also a factor in bacterial survival on dry fomites in addition to the temperature. Specifically, the experiments with S. aureus ATCC 29213 and E. coli ATCC 25922 were adjusted to humidity of 40-60% in the incubator by placing a 100-ml beaker filled with water. The humidity of the incubator without the beaker was less than 20%. Humidity was measured with a general hygrometer. Although the trend observed with altered temperature did not change, humidity affected their survival on dry surface in the opposite way. Specifically, humidity control (40-60%) weakened the thermal effect on the survival of S. aureus on the dry environment, whereas it had an inhibitory synergistic effect on the survival of E. coli. We have reflected these results in the revised manuscript with a supplementary figure (S3 Fig).

Comment 5:

“The esthetic appearance of the graphs needs to be improved, particularly be modifying to provide a more streamlined numerical representation of y axis intervals.”

According to the suggestion, we have improved and simplified the y-axis intervals of all figures in the revised manuscript. 

 

JOURNAL REQUIRMENTS

Comment 1:

“Please ensure that your manuscript meets PLOS ONE's style requirements, including those for file naming. The PLOS ONE style templates can be found at http://www.journals.plos.org/plosone/s/file?id=wjVg/PLOSOne_formatting_sample_main_body.pdf and http://www.journals.plos.org/plosone/s/file?id=ba62/PLOSOne_ formatting_sample_title_authors_affiliations.pdf”

According to the PLoS ONE guidelines, we have carefully revised our manuscript, including Heading levels, Figure Citations, Figure Captions, Reference Citations, Supporting Information Citations, Acknowledgments, References, and Supporting Information Captions. 

Comment 2: 

“Our editorial staff has assessed your submission, and we have concerns about the grammar, usage, and overall readability of the manuscript. We therefore request that you revise the text to fix the grammatical errors and improve the overall readability of the text before we send it for review. We suggest you have a fluent, preferably native, English-language speaker thoroughly copyedit your manuscript for language usage, spelling, and grammar.”

According to the suggestion, the revised manuscript has been copyedited by an English-proofreading company, Edanz (https://jp.edanzgroup.com). 

 

REPLIES FOR REVIEWER’S COMMENTS 

Comment 1:

“This manuscript addresses the effect of different temperatures on the survival on dry materials of 2 species of bacteria associated with nosocomial infections.”

According to the comment, we performed additional experiments with several bacteria (E. faecalis, S. aureus, P. aeruginosa, A. baumannii, S. marcescens and B. subtilis) and a yeast-like fungus (C. albicans). Specifically, the survival of these human pathogens on the handrail device was monitored. As a result, the prolonged survival of bacteria (E. faecalis, S. aureus, P. aeruginosa, A. baumannii, and S. marcescens) as well as E. coli, were significantly inhibited, although no effective inhibition of spore-forming bacterium (B. subtilis) was observed (Fig 6). In addition, yeast-like fungus, C. albicans, failed to survive immediately after it was added on the device (S5 Fig). We have reflected these results in the revised manuscript with a figure (Fig. 6) and supporting information (S5 Fig).

Comment 2:

“Lines 56-58 and 91-93: Even though E. coli and S. aureus are the most frequent nosocomial infection isolates, they do not represent the majority of cases. As such, microorganismos such as Enterococcus spp, P. aeruginosa, A. baumannii, Klebsiella spp, coag neg Staphilococci, Candida spp and C. difficile could be included in order to have representativeness of other Gram positive cocci, Enterobacteriaceae, non-Enterobacteriaceae Gram negative bacilli, yeasts and anaerobes. Therefore, claiming that the "thermal control of dry fomites has the potential to control bacterial survival on high-touch surfaces in hospitals, therefore reducing the spread of nosocomial infections" may sound abusive.”

As mentioned above, we have assessed if the human body core temperature can affect the survival of other bacteria (E. faecalis, S. aureus, P. aeruginosa, A. baumannii, S. marcescens and B. subtilis) and fungus (C. albicans) on dry fomites. As a result, the prolonged survival of the studied human pathogens on dry fomites were inhibited, except spore-forming bacterium, B. subtilis. Meanwhile, as pointed out by the reviewer, the number of tested human pathogens are still limited. As advised by the reviewer, we have toned down the description of the relationship of thermal controlling with the prevention of nosocomial infection throughout the revised manuscript. In particular, the descriptions related to thermal controlling with the spreading of nosocomial infections were completely deleted in the revised manuscript.

Comment 3:

“Line 99: Please consider materials and methods before results.”

According to the suggestion, we have placed the methods section above the results in the revised manuscript. The citation reference number was also revised accordingly.

Comment 4:

“Lines 123-124: Results of the study demonstrate that some strains of S aureus and E coli favor a lower temperature for surviving on dry surfaces... please consider that the spread of nosocomial infection was not addressed.”

According to the suggestion, we have properly revised the sentence. 

Thus, our findings showed that these bacteria favor a lower temperature for their survival in dry environments. Specifically, we omitted the description related to “ the spread of pathogens”.

Comment 5:

“Lines 253-255: Again, the generalisation that "warming of fomites in hospitals to skin temperature, to control the survival of human pathogenic bacteria responsible for nosocomial infections" sounds excessive.”

According to the suggestion, we have toned down the description. Specifically, we omitted that phrase and modified the sentences in the conclusion of the revised manuscript as follows:

In conclusion, we demonstrated that human pathogens (E. coli, S. aureus, E. faecalis, P. aeruginosa, A. baumannii, S. marcescens and C. albicans) responsible for nosocomial infections favor a lower temperature for their survival in dry environments. Based on this finding, we proposed a novel and simple strategy involving the warming of high-touch fomites in hospitals to body core temperature, to control the survival of human pathogenic bacteria responsible for nosocomial infections. However, this study had several limitations. As mentioned above, humidity, which can also influence bacterial survival, was not accurately controlled throughout our experiments [34-36, 48, 49], and also the range of human pathogens used for this study were limited. Further studies are needed to clarify the effect of humidity on bacterial survival under dry conditions and to confirm the reproducibility of our findings in other bacteria. Further innovations are also needed to develop handrails or doorknobs with a uniform surface temperature equivalent to that of the core human body temperature before practical application in hospital environments.

---

## [Editor Report · Decision Letter 1]

11 Dec 2019

Effect of thermal control of dry fomites on regulating the survival of human pathogenic bacteria responsible for nosocomial infections

PONE-D-19-18407R1

Dear Dr. Yamaguchi,

We are pleased to inform you that your manuscript has been judged scientifically suitable for publication and will be formally accepted for publication once it complies with all outstanding technical requirements.

With kind regards,

Thomas Byrd

Academic Editor

PLOS ONE

Additional Editor Comments (optional):

Line 52 – “Although the trends remained changed, adjusting…”. It seems as though the intention is for this to read that trends remained unchanged.  Please revise if this is correct before final version is submitted.
---

## [Editor Report · Acceptance letter]

17 Dec 2019

PONE-D-19-18407R1 

Effect of thermal control of dry fomites on regulating the survival of human pathogenic bacteria responsible for nosocomial infections 

Dear Dr. Yamaguchi:

I am pleased to inform you that your manuscript has been deemed suitable for publication in PLOS ONE. Congratulations! Your manuscript is now with our production department. 

With kind regards,

on behalf of

Dr. Thomas Byrd 

Academic Editor

PLOS ONE